# Preparation, Characterization, and Bioavailability of Host-Guest Inclusion Complex of Ginsenoside Re with Gamma-Cyclodextrin

**DOI:** 10.3390/molecules26237227

**Published:** 2021-11-29

**Authors:** Hui Li, Guolei Zhang, Wei Wang, Changbao Chen, Lili Jiao, Wei Wu

**Affiliations:** Jilin Ginseng Academy, Changchun University of Chinese Medicine, Changchun 130117, China; lihuiterrisa@163.com (H.L.); snowwinter1984@sina.com (G.Z.); 18804315443@163.com (W.W.); ccb2021@126.com (C.C.)

**Keywords:** ginsenoside Re, inclusion complex, molecule docking, bioavailability, LC-MS/MS

## Abstract

This work aimed at improving the water solubility of Ginsenoside (G)-Re by forming an inclusion complex. The solubility parameters of G-Re in alpha (α), beta (β), and gamma (γ) cyclodextrin (CD) were investigated. The phase solubility profiles were all classified as AL-type that indicated the 1:1 stoichiometric relationship with the stability constants Ks which were 22 M^−1^ (α-CD), 612 M^−1^ (β-CD), and 14,410 M^−1^ (γ-CD), respectively. Molecular docking studies confirmed the results of phase solubility with the binding energy of −4.7 (α-CD), −5.10 (β-CD), and −6.70 (γ-CD) kcal/mol, respectively. The inclusion complex (IC) of G-Re was prepared with γ-CD via the water-stirring method followed by freeze-drying. The successful preparation of IC was confirmed by powder X-ray diffraction (XRD), Fourier transform-infrared spectroscopy (FT-IR), differential scanning calorimetry (DSC), and scanning electron microscopy (SEM). In-vivo absorption studies were carried out by LC-MS/MS. Dissolution rate of G-Re was increased 9.27 times after inclusion, and the peak blood concentration was 2.7-fold higher than that of pure G-Re powder. The relative bioavailability calculated from the ratio of Area under the curve AUC_0_–_∞_ of the inclusion to pure G-Re powder was 171%. This study offers the first report that describes G-Re’s inclusion into γ-CD, and explored the inclusion complex’s mechanism at the molecular level. The results indicated that the solubility could be significantly improved as well as the bioavailability, implying γ-CD was a very suitable inclusion host for complex preparation of G-Re.

## 1. Introduction

Ginsenoside Re (G-Re), a protopanaxatriol with a tetracyclic triterpene structure, is one of the most abundant ginsenosides in *Panax ginseng*. Its content in ginseng root was more than 1% [1], and was even 2.6–4 times higher than that in ginseng leaf and root hair [2]. Studies reported that G-Re has a variety of pharmacological effects, especially on the cardiovascular system [3,4,5]: G-Re has been proven to play a role in reducing ischemic damage to cardiomyocytes. It maintains calcium transport, protective function, and mitochondrial structure in cardiomyocytes, while enhancing their antioxidant capacity [6]. Additionally, G-Re can also reduce myocardial damage and inhibit cardiac hypertrophy, indicating that G-Re may have the potential to inhibit ventricular remodeling and promote the healing process after infarction [7]. However, pharmacokinetic studies have shown that after oral administration, the tmax of G-Re was short and the bioavailability was low [8,9,10]. Due to the poor solubility and insufficient blood concentration, G-Re is not suitable for drug development [11]; therefore, improving the solubility of G-Re and thereby its bioavailability are the key points for the drug development of G-Re.

Cyclodextrin (CD) inclusion is a method to improve the dissolution of poorly soluble compounds. CD can form inclusion complexes (ICs) with hydrophobic compounds of appropriate size to improve their dissolution by virtue of the external hydrophilic group and the internal hydrophobic cavity. CDs are cyclic oligosaccharide macromolecules containing α-1 and 4-linked glucopyranose units, and they are widely used in the pharmaceutical and food industries (Figure 1) [12]. Natural cyclodextrin (CD) can be divided into α-, β-, and γ-CD that contains 6, 7 or 8 d–(+)-glucopyranose monomer linkages, respectively [13,14,15,16]. This approach has obvious advantages in improving the dissolution of the active ingredients of herbal medicine extracts through interacting with intermolecular forces such as hydrogen bond, Vander Waals force, and hydrophobic interaction, and finally changes the physical and chemical properties of the compounds [17,18]. Igami [19] prepared IC of a metabolite of ginsenoside-compound K with γ-CD, and found that γ-CD enhanced the solubility of compound K. The IC of pseudoginsenoside F11 with γ-CD prepared by Floresta [20] suggested that cyclodextrin inclusion can change the molecular structure. Additionally, when combined with β-CD, a remarkable improvement of the solubility and bioavailability of intestinal metabolite of ginseng saponin (IH901) was shown [21]. There is only one sugar residue chain in both compound-K (CK) and IH901, while there are three sugar residues in G-Re, which are connected to the C6 and C20 positions of the steroid core, respectively (Figure 1). Theoretically, the size of this molecule is too big to be included completely in the cavity of CD. So, can compounds with a relatively large molecular diameter like G-Re form IC with CD? What kind of conformation will be formed, and how much will the dissolution and bioavailability of the IC be improved?

## 2. Results and Discussion

### 2.1. Phase Solubility Studies

The inclusion approach was applied to improve the solubility and bioavailability of G-Re in this study. As shown in Figure 2 and Table 1, in order to investigate the inclusion ability of different CDs to G-Re, the phase solubility of G-Re with different concentrations of α-CD, β-CD and γ-CD was studied at 25 °C. The inclusion constants of G-Re in α-, β-, and γ-CD were obtained via calculation.

Linearity is the characteristic of an AL-type system in phase solubility which implied that the water-soluble complexes were formed in solution. In addition, the slope value was always lower than 1.0, indicating the guest (G-Re) and host (CDs) molecules constructed an inclusion compound with a molar ratio of 1:1 [22].

The apparent stability constants (Ks) were calculated from the slope of the phase solubility diagrams. The values of Ks, S0, the corresponding slopes, and the correlation coefficients of phase solubility are summarized in Figure 2 and Table 2.

Negative values of △G indicated that the complexation of G-Re and CDs could be spontaneous in aqueous solution. The Ks value of γ-CD (14410 M^−1^) was greater than that of β-CD (612 M^−^^1^) and α-CD (22 M^−^^1^) suggested that γ-CD could form more stable inclusion complexes with G-Re than α-CD and β-CD. Due to the differences of cavity’s internal space, γ-CD has the largest inclusion constant, which were 23.52 times that of β-CD and 655 times that of α-CD. Furthermore, our results showed that G-Re solubility increased linearly as the concentration of γ-CD increased, suggested a molar ratio of 1:1 between the host molecule γ-CD and guest molecule G-Re. On the other hand, Igami reported that the increasing trend of the solubility of CK with γ-CD reached the platform and gradually descended as the γ-CD was 20 mM for the 1: n complex was formed [19]. This may be due to the fact that CK only contains one glucose residue at C3, while the G-Re contains three glucose residues at C6 and C20 which would inhibit the formation of the 1: n complex.

### 2.2. Molecular Docking Simulation

In order to rationalize the mechanism of the inclusion process, molecular modeling studies were performed. The Lamarckian Genetic Algorithm method was employed for conformation searches 50 times. In auto dock, the binding energy △G was calculated according to the formula as follows [23]: △G = _(Binding Energy)_ = △G_(Intermolecular Energy)_ + △G_(Internal Energy)_ + △G_(Torsional Energy)_ + △G_(Torsional Energy)_ − △G_(Unbound Extended Energy)_. Binding energies were calculated via the Lamarckian genetic algorithm and conformations that constitute each cluster were defined by a root mean square deviation (RMSD) tolerance of 2.0. The calculated binding energies were −4.70 kcal /mol (α-CD), −5.10 kcal /mol (β-CD), and −6.70 kcal/mol (γ-CD), respectively. For the lower binding energy, the conformation formed were more stable [24]. Therefore, the structure of G-Re-γ-CD inclusion complex was more stable than the other two, which was consistent with the result of the phase solubility study.

If the distance and angle of two heteroatoms are within a certain range (generally considered to be <3.5 Å), it is considered that they can form a hydrogen bond interaction mediated by H atoms, that is, share a hydrogen atom to form a hydrogen bond [25]. The most stable conformations of G-Re-CDs ICs are shown in Figure 3. In the conformations of the three inclusion compounds, G-Re extended from the wide side to the narrow side of the cyclodextrin. However, affected by the cavity size, the binding approaches of G-Re to CDs differed. When G-Re co-existed with α-CD, the glucose residue at position C20 inserted into the cavity from wide side with six hydrogen bonds formed with the oxygen, and hydrogen atoms with the hydrogen bond lengths were (2.5, 2.8, 2.4, 2.5, 2.5, 2.3 Å) (Figure 3A). Therefore, for the complex of G-Re-α-CD, hydrogen bonding was the main force to maintain stability. When G-Re encountered β-CD, G-Re inserted its glucose residue at C6 into the cavity also from the wide side with seven hydrogen bonds formatted with the oxygen on glucose residue and hydrogen on β-CD (3.3, 3.1, 3.1, 2.9, 3.2, 2.8, 3.2 Å) (Figure 3B). The cavity size of β-CD is larger than that of α-CD, so there is more space to accommodate the guest. The β-CD provided a more stable conformation for the guest, and on the other hand, it can also provide more opportunities for the formation of hydrogen bonds. Furthermore, the theoretical calculation results showed that the binding energy of G-Re-β-CD was lower than that of α-CD, which was also consistent with the calculation results of the Ks. γ-CD is a cavity structure composed of eight D-(+)-glucopyranose monomers with the cavity size of 7.5–8.3 Å. As seen from Figure 3C, γ-CD could provide enough space to the chain of the glucose residue at C20, which passes through its cavity. Four hydrogen atoms (1.9, 2.1, 2.0, 2.4 Å) were formed in the hydroxyl group on the glucose residue at C20 and C6 that participated in the formation of hydrogen bonds from the wide side and the narrow side, respectively, which provided a force for the stability of the inclusion. However, the hydrogen bond generated by the combination of host molecule (cyclodextrin) and guest molecule (g-re) is the result of molecular dynamics simulation. A more accurate combination mode needs to be further verified by NMR experiments.

Molecular docking studies were adopted to elaborate the complex mechanism of host and guest molecules [26]. The structures of host molecule-CDs were fairly rigid due to intra-molecular hydrogen bonds between the O-2H and the O-3H of adjacent glucose units [27]. Therefore, a semi-flexible docking form was adopted for the simulation of G-Re and CDs. The docking calculation results provided an interpretation of the solubility test in the formation of inclusion compounds between G-Re and CDs at the molecular level. From a structural point of view, this stable complex of γ-CD and G-Re was attributed to the larger cavity volume of γ-CD and the more stable spatial conformation formed by the guest and the host. Docking calculation results also provided theoretical support for the 1:1 molar ratio inclusion complex inferred by the phase solubility. As for the molecule size, only the glycoside residue of G-Re can extend or partially extend into the cavity. The most stable conformation of the non-inclusion structure was parallel to the wide side of the CD. In this way, G-Re does not have enough space to combine with other cyclodextrin molecules. Therefore, only a 1:1 molar ratio inclusion complex could be conformed. The calculated results of Docking further confirmed the calculated solubility constant from a microscopic point of view. According to the docking calculation, the differences in cavity sizes might be the reason why CDs showed different inclusion constants. γ-CD could include the whole sugar residue at C20 through the cavity, which was conducive to the formation of hydrogen bonds between the hydroxyl groups on both sides of the wide and small segments with CDs.

### 2.3. G-Re Content Test

An HPLC-based method was applied for the actual G-Re content detection. Equal amounts of G-Re and γ-CD were shaken at 30 °C for 30 min to prepare a physical mixture for comparison. After measurement, the G-Re content of the physical mixture was 45.03% ± 2.3% (weight ratio ± SD). The content of G-Re in the inclusion complex was 40.17% ± 1.8%.

### 2.4. Physicochemical Characterization for the Confirmation of Inclusion Complex

#### 2.4.1. Fourier-Transform Infrared (FT-IR)

FT-IR spectroscopy is a tool to verify the IC formation, which relies on the detection of the changes of host and guest absorption peaks’ in shape, intensity, and displacement of the absorption peaks [28,29,30]. The infrared spectra of IC and PM were basically the same in peak shape and wave numbers. If there were factors affecting the characteristic absorption frequency of functional groups in the inclusion complex, such as ring finishing, charge transfer, hydrogen bond formation, and so on, the characteristic absorption of functional groups would change slightly before and after inclusion. In this study, FT-IR spectra of γ-CD, G-Re, G-Re-γ-CD PM, and G-Re-γ-CD IC were obtained. Interestingly, as depicted in Figure 4, the FT-IR spectrum of G-Re-γ-CD IC presented similar vibrational patterns to those of individual γ-CD and G-Re, suggesting the successful complexation of the G-Re in γ-CD hydrophobic cavity. The FT-IR spectrum of G-Re (Figure 4A) characterized with several strong and broad absorption bands appeared at 3419.70 cm^−1^ (stretching vibration of –OH), 2931.19 cm^−1^ (stretching vibration of –CH and –CH_2_), 1640.81 cm^−1^ (stretching vibration of C=C), 1074.99 cm^−1^, and 1046.51 cm^−1^ (stretching vibration of C–O). The evident absorption peak of γ-CD (Figure 4B) at 3388.41 cm^−1^ is attributed to the –OH stretching vibration; the 2927.00 cm^−1^ peak belonged to the stretching vibration of –CH and –CH_2_; the peaks at 1028.47, 1079.86, and 1157.15 cm^−1^ characterized the stretching vibration of C-O. After the formation of the G-Re-γ-CD, the characteristic absorption signals of G-Re at 3419.70 cm^−1^ and 2931.97 cm^−1^ were shifted to 3395.96 cm^−1^ and 2928.93 cm^−1^ (Figure 4D). The absorption peaks of G-Re at 1074.99 cm^−1^ and 1045.36 cm^−1^ transferred to 1079.86 cm^−1^ and 1028.47 cm^−1^. In addition, the relative intensity and peak shape from the FT-IR spectra of the G-Re-γ-CD complex in the range of 500–1500 cm^−1^ were fairly different from that of G-Re. These results suggested that G-Re entered the hydrophobic interior of γ-CD.

#### 2.4.2. X-ray Diffraction (XRD)

The powder X-ray-diffraction patterns of each sample are shown in Figure 5. The characteristic diffraction peaks of each sample mainly appeared in the range of 2θ < 30° [31]. The diffractgrams of G-Re (Figure 5A) and γ-CD (Figure 5B) exhibited a series of intense peaks, which were indicators of their crystalline characters: G-Re presented characteristic diffraction peaks at 9.1°, 11.05°, 14.3°, 15.85°, 17.55°, 19.45°, 24.5°, 25.35°, 27.45°, and 28.75°/2θ, and γ-CD presented characteristic diffraction peaks at 5.25°, 6.3°, 9.85°, 10.75°, 11.7°, 12.55°, 14.05°, 15.50°, 16.55°, and 18.95°/2θ. The X-ray diffraction pattern of PM showed the superposition of the spectra of each single component indicating that no new structure was formed during the Physical mixing process (Figure 5C). In contrast, only a large diffraction ring pattern could be observed in the G-Re-γ-CD IC with fewer broader and less intense peaks, which was in a diffuse shape (Figure 5D) with the disappearance of the prominent crystalline peak of G-Re situated at 9.10° and 19.45° (2θ). For the amorphous state, it lacks a long-range ordered three-dimensional molecular structure [32]. These results showed that G-Re was deposited in γ-CD at amorphous state, which could be beneficial to the dissolution of G-Re.

#### 2.4.3. Differential Scanning Calorimetry (DSC)

Thermal analysis spectrum can confirm the formation of inclusion complex from another respect. Crystals usually have a distinct melting point (Tm), yet the amorphous form often takes the glass transition temperature (Tg) as the characteristic thermodynamic parameter. However, when the guest molecule was embedded in the cavity or crystal lattice of the host molecule, the melting point, boiling point, or sublimation point of the guest molecule would usually shift to a different temperature or disappear [33]. Four endothermic peaks were observed on the DSC curve of G-Re with dehydration points at 98.5 °C and 112.8 °C, and a melting point at 248.8 °C (Figure 6A). After γ-CD was heated, it produced a broad endothermic peak at 98.66 °C, which might be the result of water loss (Figure 6B). The thermal curve of the G-Re-γ-CD complex showed a completely disappeared G-Re endothermic peak (Figure 6D), indicating the formation of an amorphous solid dispersion or the molecular encapsulation cavity of the drug in γ-CD.

#### 2.4.4. Scanning Electron Microscopy (SEM)

SEM is a qualitative method used to study the morphology of substances [34,35]. The SEM images of G-Re (Figure 7A), γ-CD (Figure 7B), G-Re-γ-CD physical mixture (Figure 7C), and G-Re-γ-CD inclusion complex (Figure 7D) are shown. The pure G-Re appeared as irregular blocky particles and γ-CDs were smooth and blocky particles with irregular shape. The inclusion compound particles, which were small and irregular, showed completely different characteristics in shape and size compared with those of G-Re and γ-CD, suggesting the successful formation of the inclusion complex. Besides, the SEM image showed that the shape and size of the G-Re-γ-CD complex were quite different from those of G-Re and γ-CD particles. Together, these results confirmed the formation of the G-Re-γ-CD inclusion complex.

### 2.5. Dissolution Test

The dissolution of G-Re, the physical mixture, and the G-Re-γ-CD inclusion complex in water were detected within an hour with a time interval of 10 min. The dissolution profiles are illustrated in Figure 8. The dissolution rate of G-Re began to reach equilibrium at 30 min. At this time, the dissolution rate of G-Re powder, physical mixture, and inclusion complex were 6.73%, 11.15%, and 79.13%, respectively. Compared with the solid powder, the dissolution rate of the physical mixture increased 1.65-fold, indicating that γ-CD enhanced the solubility on G-Re. This effect may be related to the surface-active volatilization of γ-CD in the aqueous solution. Yet, the dissolution rate of G-Re increased 9.27 times after inclusion. This result implied that the inclusion effect of γ-CD accounted for the increased solubility of G-Re (in the inclusion complex) rather than surface-active volatilization.

### 2.6. Bioavailability Study with LC-MS/MS

#### 2.6.1. Method Validation

Typical chromatograms of G-Re and internal standard (IS) are shown in Figure 9. The retention times of G-Re and IS were 2.1 and 7.1 min, respectively. Control samples collected before drug administration showed no peaks which interfered with G-Re or IS signals. The average regression equation of G-Re in plasma was Y = 0.2334X + 1.5645 (r = 0.9936, n = 3). Where, Y was the peak area ratio of each compound to IS, and X was the concentration. It was shown that there was significant linearity over the concentration range of 1–100 ng/mL. The LOD with a signal-to-noise ratio (S/N) > 3 was 0.0068 ng/mL, and the LOQ with a signal-to-noise ratio (S/N) > 10 was 0.0228 ng/mL. The intra- and inter-assays precisions ranged from 4.7% to 11.4% and 4.6% to 19.0%. The rat plasma used in experiments was available for 30 days at −80 °C and three freeze-thaw cycles at –20 °C. It was also stable during the 24 h period of storage in the autosampler. Overall, plasma samples had good stability during the analysis and were able to meet the requirements of pharmacokinetic studies.

#### 2.6.2. Pharmacokinetic Studies

After oral administration of G-Re powder and inclusion compound (100 mg/kg), the plasma concentrations were determined by LC-MS/MS. The concentration-time curves are shown in Figure 10. DAS 2.0.1 pharmacokinetic software was used for the calculation of pharmacokinetic parameters (Table 3). According to the Akaikee’s information criterion (AIC), the metabolism of G-Re in rats conformed to the characteristics of the two-compartment model with a weight coefficient of 1, 1, 1/C2. The peak concentration (Cmax) of the inclusion complex was 1.87 times higher than that of G-Re powder, and the time to reach Cmax was shorter than that of G-Re powder. AUC_0–t_ increased from 113.031 ± 44.6 ng·h/mL to 175.426 ± 67.426 ng·h/mL. The relative bioavailability of G-Re-γ-CD was calculated according to the mentioned equation, which was 171%. The enhanced AUC and relative bioavailability suggested a higher absorption rate of G-Re-γ-CD inclusion complex than G-Re powder. Regarding pharmacokinetic parameters, there were significant differences between the γ-CD complex and G-Re. Although the formation of the complex enhanced the bioavailability of G-Re, it was not as significant as the increase in solubility. As it was reported that G-Re would be further metabolized in the gastro-intestinal tract after oral administration, poor membrane permeability might be another factor limiting the bioavailability of G-Re [22]. Thus, the γ-CD complexes might be useful as a fast-dissolving form of G-Re when compared with the pure G-Re powder.

Compared with the group administered G-Re powder, the Tmax of G-Re-γ-CD IC was shorter, the AUC_0–24h_ increased, and the bioavailability was improved. On the one hand, the increase in G-Re content in the body at molecular level led to more G-Re molecules to penetrate the mucus layer and the immobile water layer. Additionally, the contact surface of the G-Re cell membrane increased. This result suggested that increased solubility of G-Re could significantly enhance its blood access. Moreover, the oil-water partition coefficient of G-Re-γ-CD IC was larger than that of G-Re powder [36,37]. It was speculated that G-Re-γ-CD maintained the coexistence balance of inclusion compound and non-inclusion complex, and played a transfer role at the water-biological phase interface. This increased the amount of G-Re permeated through the bio-film, thereby causing improved bioavailability. The increased G-Re content in the blood took longer to clear, so T1/2 of G-Re-γ-CD IC was prolonged. Additionally, related studies have shown that cyclodextrin and its derivatives can deplete cholesterol on cell membranes and change their structure and permeability, thus inhibiting the activity of P-glycoprotein which is closely related to the bioavailability of oral drugs [38]. These properties of dextrin increase the probability of the drug entering the intestinal epithelial cells, prolonging the drug MRT(h) and T1/2(h). The inclusion compound can also improve the stability of the drug in body fluids and increase its absorption [39]. These findings suggest that the preparation of G-Re into G-Re-γ-CD inclusion compound can improve its oral bioavailability. The studies mentioned above have important reference significance for the development of G-Re and related pharmaceutical dosage forms.

## 3. Materials and Methods

### 3.1. Chemicals

G-Re (purity > 98%) was provided by Yuanye Bio-Technology Co., Ltd. (Shanghai, China). α-CD, β-CD, and γ-CD also came from Yuanye Bio-Technology Co., Ltd. (Shanghai, China). Solvents used in the G-Re analysis were HPLC grade and purchased from Tedia (Fairfield, OH, USA).

### 3.2. Phase Solubility Test

Phase solubility studies were performed in water in accordance with the method formerly stated by Higuchi and Connors [20]. In brief, an excess amount of G-Re was supplemented into 10 mL Eppendorf tubes with aqueous solution, including various concentrations of α-CD, β-CD, and γ-CD, respectively. Suspensions were vigorously shaken in 25 °C water bath for a week. After equilibrium was reached, the solutions were centrifuged for 5 min at 4000 rpm (1500 g). Meanwhile, a 0.22 μm membrane was used to further filter the supernatant. The filtrate was diluted and HPLC was employed to determine G-Re content.

The phase solubility curves were obtained with the regression of the concentrations of cyclodextrin (mol L^−1^) as X-axis and the concentration of G-Re (mol L^−1^) as Y-axis. The value of apparent stability constant (*Ks*) was defined by using the straight-line portion in the phase-solubility to the equation as follows:$$K_{s}=\frac{\mathrm{slope}}{S_{0}(1-\mathrm{slope})}$$

S_0_ shows the intrinsic solubility of G-Re without CDs, and the Gibbs free energy change (△G) in the complexion process could be calculated as follows: ΔG = −RTln *Ks*

In this formula, R can be seen as the general gas constant, which equaled 8.314 J/mol K, and T can be considered as the experimental operating temperature.

### 3.3. Molecular Docking Simulation

The PDB ID of α-CD, β-CD, and γ-CD were 2 zym, 2 zyn, and 2 zyk, respectively. The molecular models were obtained from the Protein Data Bank (PDB) at www.rcsb.org. The combined protein molecules were removed with UCSF Chimera 1.15.rc (www.rbvi.ucsf.edu/chimera), and the molecules were saved as “.pdb” format. The structure of G-Re was drawn with Chembiodraw Ultra 13.0 (CambridgeSoft, Cambridge, MA, USA). In addition, the three-dimensional (3D) structures of G-Re was generated by ChemBio 3D Ultra 13.0 (CambridgeSoft, Cambridge, MA, USA), and then minimized energy with MM2.

As for docking G-Re into CD cavity, Auto Dock Vina software (Version 1.5.6, Sep _17_14, TSRI, California, CA, USA) was employed. 

Guest G-Re was prepared using “ligand” and all rotatable bonds in its structures were set rotatable. After removing of crystallographic water molecules, host α-CD, β-CD, and γ-CD molecules were used as the receptor. A semi-flexible docking method was employed for docking. The box sizes of CDs grids were set as 3.2 nm × 3.0 nm × 2.6 nm for α-CD, 3.2 nm × 3.2 nm × 3.0 nm for β-CD, and 3.6 nm × 4.4 nm × 2.8 nm for γ-CD. The grid spacing was 0.375 nm by default, and then the grid points were calculated. The maximum number of evaluations was 2.5 × 10^5^. The Lamarckian genetic algorithm was used to perform 100 conformation searches to discuss the inclusion of G-Re-CDs.

### 3.4. Preparation and Optimization of IC

The IC was obtained with the previously described methods with the following modifications. G-Re (50 mM) was dissolved in ethanol and added into γ-CD aqueous solution with a 1:1 molar ratio. After that, the mixture was stirred thoroughly at 25 °C for 12 h. All parameters were optimized through orthogonal experiments. Filtration was performed to remove undissolved G-Re with 0.22 μm membrane filters. Then, the filtrate was frozen at −80 °C (MDF-U54V, Panasonic Inc., Osaka, Japan). Later, the filtrate was lyophilized in the freeze-dryer (FDU-2100, Rikakikai Co., Ltd., Tokyo, Japan) for 2 days.

A physical mixture (PM) refers to a mixture prepared by physically mixing host molecules and guest molecules. Equal amounts of G-Re and γ-CD were shaken at 30 °C for 30 min to prepare a physical mixture (PM) for comparison.

### 3.5. Dissolution Test

G-Re, G-Re-γ-CD physical mixtures, and the G-Re-γ-CD ICs (20 mg as Re) were inserted into a basket in a automatic dissolution testing system (RC806D, Tianjin Tianda Technology Co.,Ltd, Tianjin, China) containing 200 mL of water according to the dissolution method (Chp. 2020 Vol II Addium XC rotatory-basket method). The speed was set at 100 rpm, and the temperature was 37 ± 0.5 °C. Then an equal amount of solution (1 mL) needed to be taken at timepoints of 5, 15, 30, 45, and 60 min; at the same time, 1 mL solution should be replaced. As described below, the determination of G-Re concentration by 205 nm high-performance liquid chromatography was feasible. The G-Re concentration was detected with the same HPLC method applied in the phase solubility test.

### 3.6. Confirmation of IC

#### 3.6.1. Fourier Transform-Infrared Spectroscopy(FT-IR)

A spectrometer (Model: Nicolet iS5, Thermo Fisher Scientific, USA) was used for infrared spectroscopy detection. The sample was ground and mixed with potassium bromide (KBr) to form an infrared transparent matrix and diluted to 1%. The scanning range was 4000 cm^−1^–500 cm^−1^, and data were collected with spectroscopy software 5.0.2 (Perkin Elmer Corporation, USA).

#### 3.6.2. Differential Scanning Calorimetry (DSC)

A thermal analyzer (model: MDSC2910, New Castle, DE, USA) on a differential scanning calorimeter (TA Instrument, New Castle, DE, USA) was used to realize the thermal analysis of the sample. In this process, alumina was used as a reference, the scanning range was set 25–300 °C, and the heating rate was increased by 10 °C per minute.

#### 3.6.3. X-ray Diffractometry (XRD)

The X-ray diffraction pattern of the powder was investigated with an instrument (model: TD 3500 XRD, TongDa Ltd., Liaoning, China) equipped with high-temperature accessories. The test conditions were Cu-Ka radiation, tube voltage 50 kV, tube current 200 mA, step length 0.02°, and samples were analyzed in the range of 2θ(θ)3–50°. The graphite monochromatic diffracted beam was monochromatic.

#### 3.6.4. Scanning Electron Microscopy (SEM)

This method was utilized to investigate and photograph the shape and surface characteristics of binary complexes and raw materials (Model: JSM-IT 100, SEM Instrument, JEOL Ltd., Tokyo, Japan).

#### 3.6.5. HPLC Analysis of Re Levels

The contents of G-Re from the dissolution sample and phase solubility sample were determined with the help of HPLC on the C18 column (Ascentis^®^ Express, 10 cm × 2.1 mm, 2.6 μm, Sigma, USA), which was installed an Agilent system. This system consisted of a micro degasser, an auto plate sampler, a system binary pump, as well as a thermostatically controlled column apartment at 30 °C. Samples were separated under elution conditions at A (acetonitrile) and B (water) = 19: 81 (*v/v*). Then, the flow rate was set at 0.5 mL per minute and the injection volume at 5 μL, while the G-Re was detected spectrophotometrically at 205 nm.

### 3.7. Bioavailability Studies

#### 3.7.1. G-Re Administration and In Vivo Analysis

The animal studies carried out in this study strictly followed the instructions of the Animal Care and Use Committee of the Changchun university of Chinese medicine (Approval Number: 20190030). Adult male Sprague-Dawley rats (200–220 g; Yisi Experimental Animal Technology Co., Ltd., Changchun, China Approved number: SCXK (JI) 2018-0007) were used for the pharmacokinetic studies. All rats were not allowed to be fed for 24 h before the oral dosing and likewise after the drug administration. Besides, all of them needed to be brought up under a controlled temperature ranging from 20 to 24 °C with a 12-h light/dark cycle.

G-Re powder and G-Re-γ-CD complex were suspended in double-distilled water prior to use at 100 mg/kg. Blood (300 μL) coming from the orbital venous plexus was be collected in the heparinized tube at 0.083, 0.25, 0.5, 1, 2, 3, 4, 6, 9, 10, 12, and 24 h, respectively. The blood samples were centrifuged for 10 min at 4000 rpm to harvest the plasma sample (100 μL). Then, the G-Rf (10 μL, 10 ng/mL) was added into each tube rapidly to work as the inherent criterion. After that, a mixed 0.5 mL methanol-acetonitrile (60:40, *v/v*) should be added for precipitating proteins and extracting interested compounds. The samples were then rotated for 3 min and centrifuged at 15,000×grams for 10 min. By separating and evaporating, the supernatant was obtained on the condition that the nitrogen flow dried gently at 40 °C. Additionally, the residue was reconstituted by vortexing for 1 min with 0.2 mL methanol-water (80:20, *v/v*). Furthermore, the supernatant was transferred into auto-sampler vials using 5 μL aliquot to inject to the LC-MS/MS system.

#### 3.7.2. Sample Detection with LC-MS/MS

This method could be carried out by a combination of a Thermo TSQ Endura triple-quadrupole tandem mass spectrometer (Thermo Fisher Inc., Waltham, USA), which is a method named electrospray ionization (ESI), and an Ultimate 3000 LC system which is composed of a vacuum degasser, a quaternary pump, as well as an auto-sampler. Xcalibur 2.2 (SP1.48) software was used to acquire and analyze the data (Thermo Fisher Inc., Waltham, USA). After that, the separation of chemical materials was conducted by using the reverse-phase Accucore C18 analytical column (100 mm × 2.1 mm, 2.6 μm, Thermo Fisher Inc., Waltham, USA). This column was available for the linear gradient elution by using the mobile phase, which consisted of water embracing 0.1% formic acid (A) and acetonitrile (B). Steps are described as follows: 0 min (A 75%–B 25%)–5.0 min (A 70%–B 30%)–8 min (A 68%–B 32%)–9 min (A 64%–B 36%)–16 min (A 63%–B 37%)–16.8 min (A 52%–B 48%)–17.8 min (A 30%–B 70%)–18.8 min (A 10%–B 90%)–20.8 min (A 75%–B 25%)–25 min (A 75%–B 25%). The flow rate should be 0.2 mL per minute, and the volume of injection needed to be 5 μL. This process needed to be performed using the ESI negative ionization and multiple reactions monitoring (MRM) mode. Furthermore, the parameters of spray voltage were 3 kV, the parameters of capillary temperature was 325 °C, and the pressure of nitrogen sheath gas and auxiliary gas were 30 arb and 10 arb. The formic acid adduct [M+HCOO]- is the dominant parent ion of products. Besides, collision energy values of G-Re and IS were 55 eV and 45 eV, respectively. Finally, the ion transition situations for multiple reactions monitoring (MRM) were *m/z* 945.5→637.3 and 637.3→475.3 for G-Re, *m/z* 799.5→637.3 and 637.3→475.3 for IS.

#### 3.7.3. Pharmacokinetic Analysis

After oral administration, non-compartment approaches could be utilized to calculate G-Re’s t pharmacokinetic parameters. The part under the plasma concentration-time curve increasing from 0 to infinity (AUC_0__→∞_) could be computed by the equation AUC_0__→∞_ = AUC_t_ + C_t/k_. In this equation, C_t_ could be seen as the last concentration which could be quantized, and the calculation of k could be conducted by a straight-line’s slope in plasma disappearance’s terminal phase. In addition, the calculation of half-life of terminal phase (t_1/2_) needed to be performed by using t_1/2_ = 0.693/k. Furthermore, with the help of linear trapezoidal approximation, it was easy to calculate the area below the plasma concentration-time curve increasing from 0 to the time of the last quantifiable concentration (AUC_t_). The calculation of bioavailability value for G-Re’s oral doses needed to be conducted according to the AUC’s dose-adjusted ratio.

The data expressed as mean ± standard deviation (SD) were processed using the Drug and Statistics (DAS 2.0) software (Medical College of Wannan, China) under the two-compartment model to calculate the following pharmacokinetic parameters: Cmax (peak plasma concentration) after oral administration, Tmax (time to peak concentration), AUC_0-24 h_ (calculate the area under the curve based on the trapezoid between 0 and 24 h), AUC_0__→∞_ (the area under the curve is extrapolated to infinity), the MRT (mean retention time), and F (relative bioavailability). Average values were also analyzed.

## 4. Conclusions

The host-guest supramolecular interactions between α-CD, β-CD, γ-CD, and G-Re were investigated in this manuscript. All experiments confirmed that G-Re and γ-CD formed a 1:1 inclusion compound. Molecular docking studies elaborated the complex mechanism. As a protopanaxatriol, G-Re can only be partially embedded in the cavity of γ-CD due to the restriction of sugar chains. Interestingly, even if a partial IC was formed, the G-Re-γ-CD complex showed an increase in dissolution rate and bioavailability. Thus, γ-CD played a significant role in the enhancement of G-Re’s dissolution and bioavailability.

An inclusion complex with G-Re and γ-CD at a molar ratio of 1:1 was further prepared. The characteristics of G-Re were different from those in the physical mixture and inclusion compound. The successful preparation of the G-Re-γ-CD IC was confirmed by FT-IR, DSC, XRD, and SEM. The dissolution rate of G-Re from the inclusion was greatly improved. The G-Re amount released from IC was 10.27-times greater than that of G-Re powder. Therefore, the dissolution rate of G-Re could be greatly improved with γ-CD inclusion. In vivo studies after oral administration demonstrated that the AUC value of γ-CD IC was higher than that of pure G-Re. After oral administration of G-Re-γ-CD IC at a dose of 100 mg/kg, the peak concentrations (Cmax) and the bioavailability of G-Re were 1.71-times higher than the pure G-Re powder. In conclusion, γ-CD is a promising inclusion host for the complex preparation of G-Re.

## Figures and Tables

**Figure 1 molecules-26-07227-f001:**
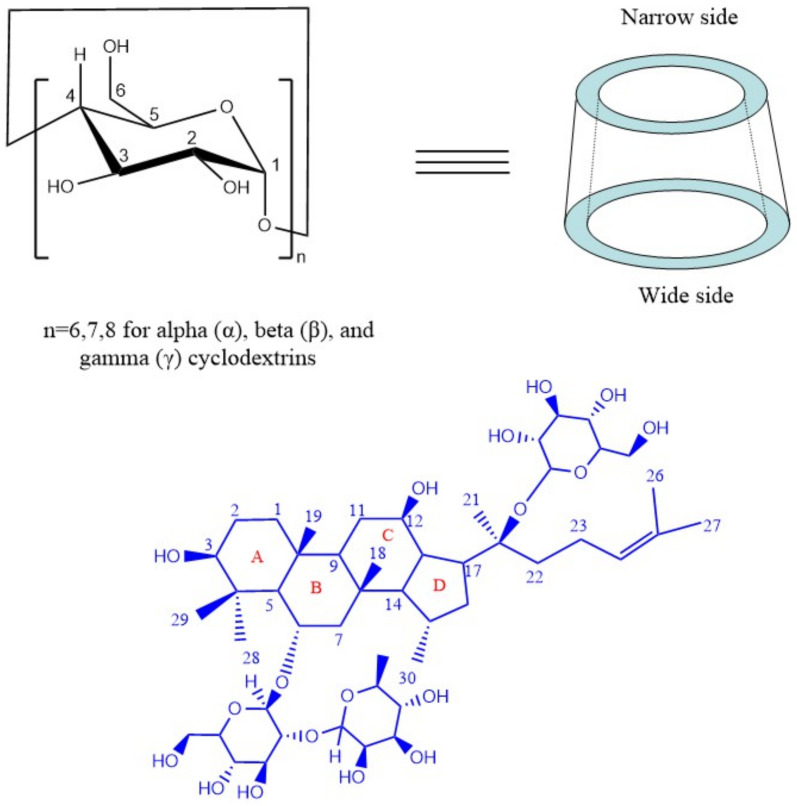
Structures of host (α-CD, β-CD, γ-CD) and guest (G-Re).

**Figure 2 molecules-26-07227-f002:**
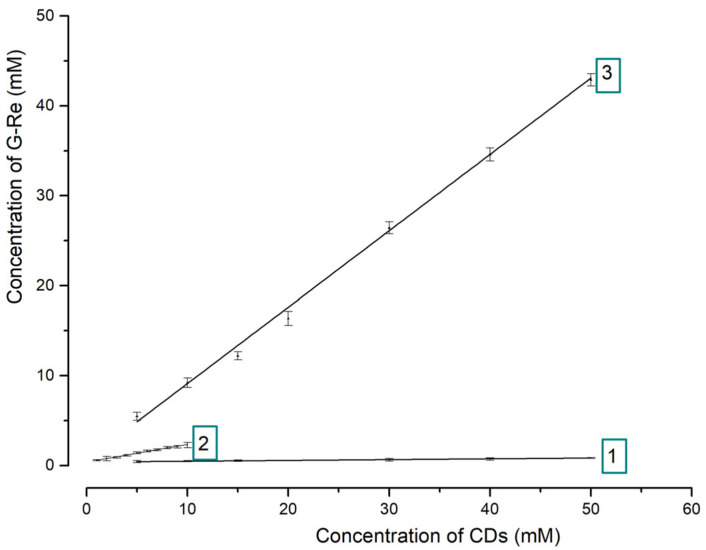
Phase solubility curves of G-Re in CDs: “1” is for α-CD; “2” is for β-CD; “3” is for γ-CD. Phase solubility curves of G-Re with different CDs α-CD, β-CD, and γ-CD were labeled 1, 2 and 3, respectively.

**Figure 3 molecules-26-07227-f003:**
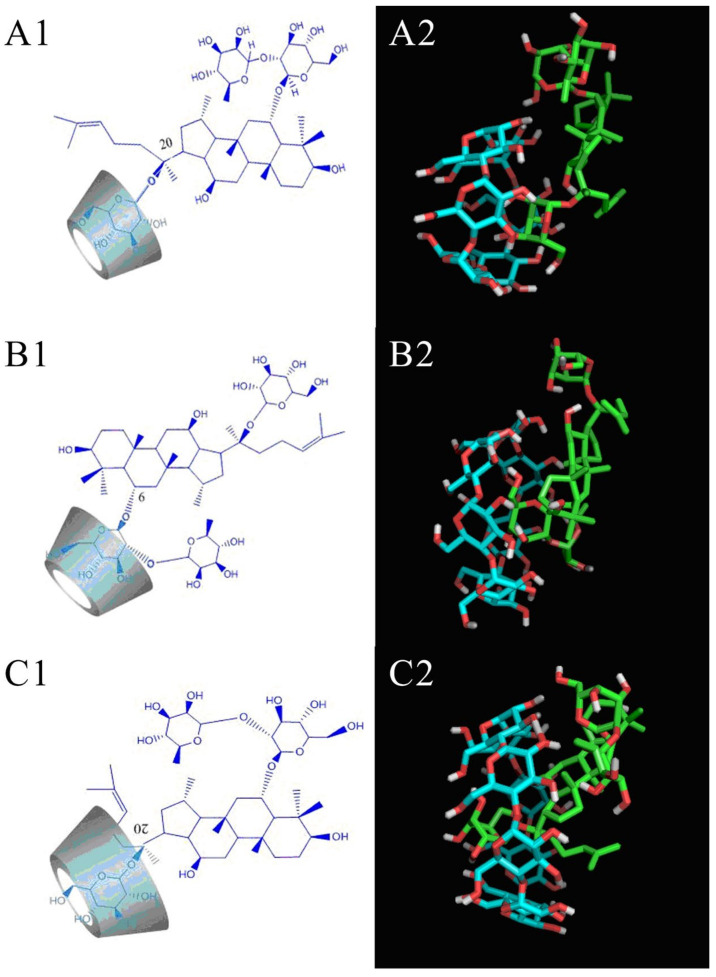
Molecular docking conformation diagrams of G-Re with CDs: (**A**: IC of G-Re-α-CD; **B**: IC of G-Re-β-CD; **C**: IC of G-Re-γ-CD). 1: Schematic diagram of the structure of the inclusion position; 2: images of the inclusion complex from Pymol software (Pymol^TM^,1.7.X). The blue sticks represented the host molecule (CDs), the green sticks represented the guest molecule (G-Re), the red part represented the oxygen atoms, and the white part represented the hydrogen atoms.

**Figure 4 molecules-26-07227-f004:**
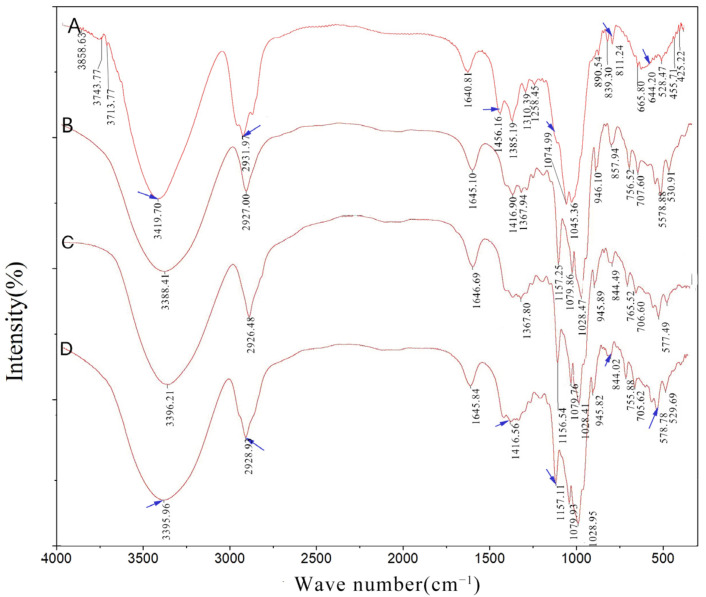
FR-IR spectra of G-Re (**A**), γ-CD (**B**), G-Re-γ-CD PM (**C**), and G-Re-γ-CD IC (**D**).

**Figure 5 molecules-26-07227-f005:**
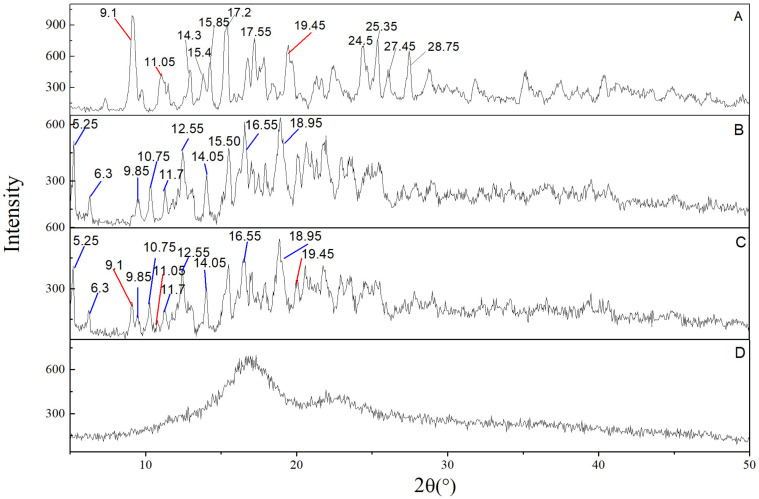
X-ray diffractograms of G-Re (**A**), γ-CD (**B**), G-Re-γ-CD PM (**C**), and G-Re-γ-CD IC (**D**). Red lines point to the characteristic peak of G-Re and blue lines point to the characteristic peak of γ-CD.

**Figure 6 molecules-26-07227-f006:**
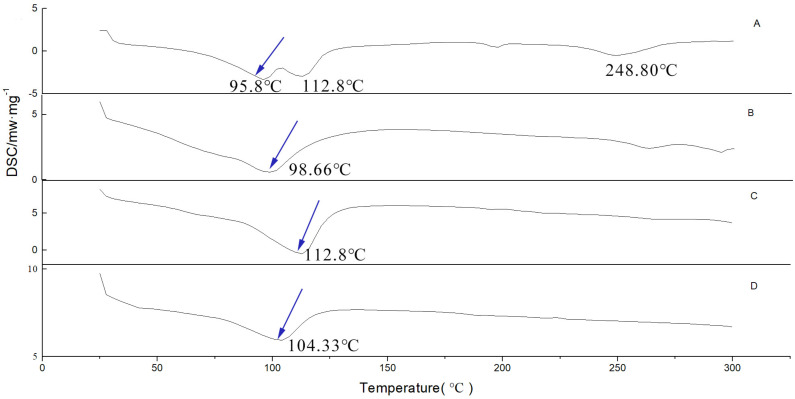
DSC thermograms of G-Re (**A**), γ-CD (**B**), G-Re-γ-CD PM (**C**), and G-Re-γ-CD IC (**D**).

**Figure 7 molecules-26-07227-f007:**
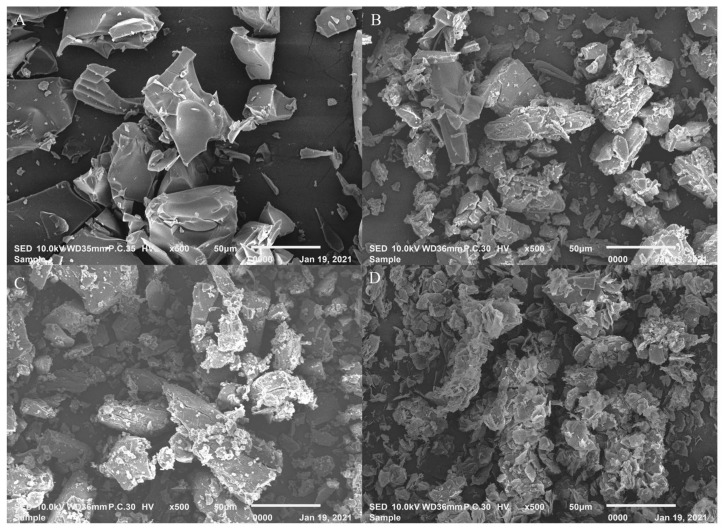
Scanning electro microphotographs of G-Re (**A**), γ-CD (**B**), G-Re-γ-CD PM (**C**), and G-Re-γ-CD IC (**D**).

**Figure 8 molecules-26-07227-f008:**
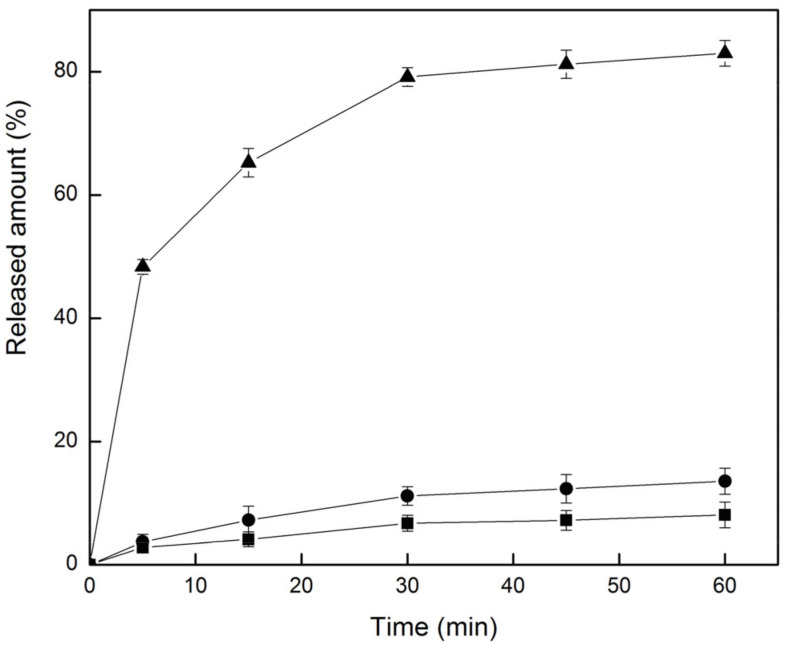
Dissolution profiles of G-Re (■), G-Re-γ-CD PM (●), and G-Re-γ-CD IC (▲) in water. Each point represents the mean ± SD. (n = 3).

**Figure 9 molecules-26-07227-f009:**
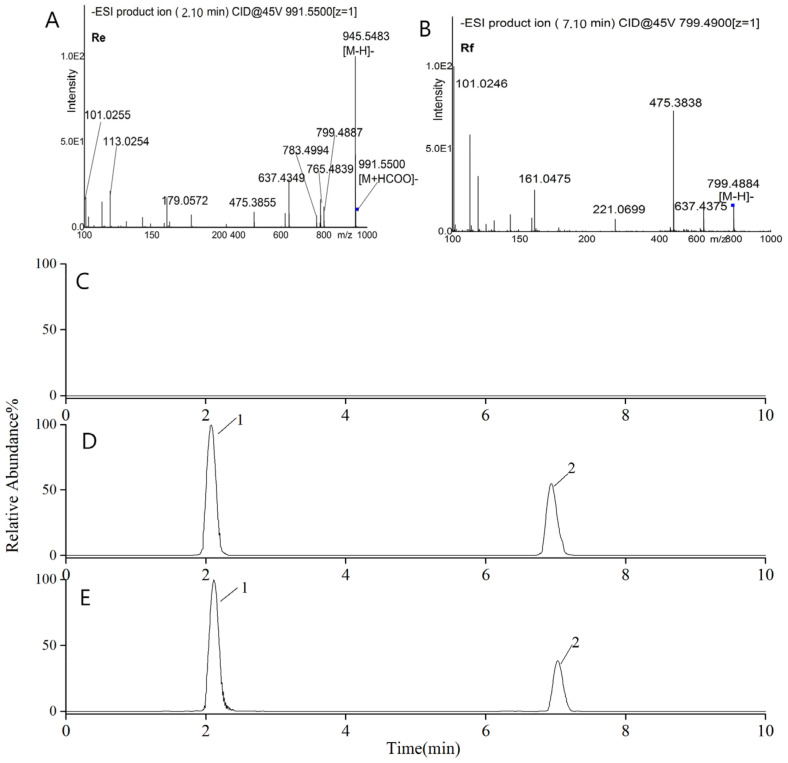
Tandem Mass spectra of G-Re (**A**), IS (**B**), and LC-MS/MS chromatograms of blank plasma (**C**), plasma spiked with G-Re and IS (**D**), and plasma sample after oral administration (**E**) respectively. Peak “1” represented G-Re, “2” represented IS. G-Re was detected at m/z 945.5→637.3 and 637.3→475.3 using MRM.

**Figure 10 molecules-26-07227-f010:**
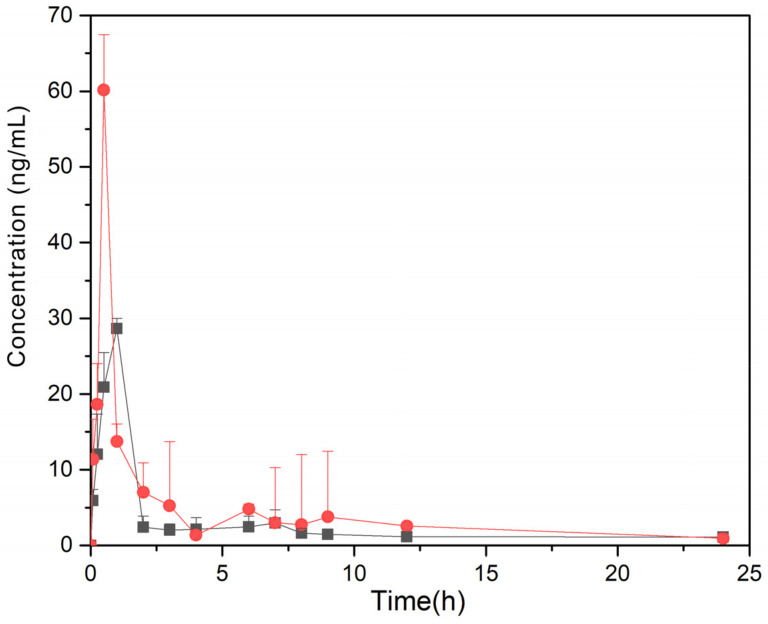
Plasma concentration-time curves after oral administration of G-Re powder (■) and G-Re-γ-CD IC (●).

**Table 1 molecules-26-07227-t001:** Phase solubility parameters of G-Re in different CDs in water.

CD	S_0_ (×10^−3^M)	Slope	Ks (×10^3^M^−1^)	1/Slope	R^2^	△→G (kJ/mol)
α-CD	0.392	0.0086	0.022	116.28	0.9938	−7.659
β-CD	0.391	0.1935	0.612	5.16	0.9933	−15.889
γ-CD	0.391	0.8494	14.41	1.18	0.9974	−23.736

**Table 2 molecules-26-07227-t002:** The hydrogen bond formations between the guest (CDs) and the host molecule (G-Re).

Host	Side	Insertion Position	Affinity Energy (kcal /mol)	Bond Numbers	O-O Intermolecular Distance (Å)
α-CD	wide	Glu-C20	−4.70	6	2.5, 2.8, 2.4, 2.5, 2.5, 2.3
β-CD	wide	Glu-C6	−5.10	7	3.3, 3.1, 3.1, 2.9, 3.2, 2.8, 3.2
γ-CD	wide	Glu-C20	−6.70	4	1.9, 2.1, 2.0, 2.4

“side”: the wide or the narrow side; “Insertion position”: the part included in the host molecule. Glc-C20 means glucose residue at C20.

**Table 3 molecules-26-07227-t003:** Main pharmacokinetic parameters of G-Re and G-Re-γ-CD IC after single oral administration in rats (mean ± SD, n = 6).

		Parameter Values	
Parameters	Measuring Unit	G-Re	G-Re-γ-CD
Cmax	ng/mL	32.237 ± 8.152	60.176 ± 10.296 *
Tmax	h	1.023 ± 0.036	0.509 ± 0.042 *
T1/2	h	12.169 ± 1.557	14.164 ± 8.060 *
AUC0–24h	ng·h/mL	113.031 ± 44.6	175.426 ± 67.426 *
AUC_0_–∞	ng·h/mL	129.907 ± 54.1	219.945 ± 127.918 *
MRT	h	5.147 ± 0.968	6.976 ± 1.465 *
CLt	L/h	2.401 ± 0.52	1.370 ± 0.28
Vdss	L/kg	13.691 ± 3.74	8.135 ± 1.47 *
F	%	—	171.395 ± 48.89 *

Note: * *p* < 0.05, vs. G-Re.

## Data Availability

All the data used in this study are available within this article. Further inquiries can be directed to the authors.

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
