# Peer review of "Preparation, Characterization, and Bioavailability of Host-Guest Inclusion Complex of Ginsenoside Re with Gamma-Cyclodextrin"

_molecules, 2021, doi:10.3390/molecules26237227_

Round 1

Reviewer 1 Report

The manuscript (MS) reports structures and thermodynamics of the inclusion complexes of a-, b-, and g-cyclodextrins (CDs) with ginsenoside Re (G-Re) based on phase solubility and molecular docking studies. The most stable G-Re-g-CD inclusion complex was then prepared, characterized (FTIR, DSC, XRD and SEM) and subject to bioavailability test. Although the overall work is interesting to general audiences, the MS is not well prepared, and it deserves major revisions as listed below before reconsideration for publication in Molecules.

  1. Results and discussion require further clarification and reverification.
  • The complex binding energies from molecular docking calculation need to be rechecked and their derivation should be given; the value of G-Re-g-CD complex (6.70 kcal/mol) is highest compared to other two complexes (-4.7 and -5.1 kcal/mol), which is in contrast to the order of Ks of the three complexes. However, the authors conclude that both binding energies and stability constants have the same tendency. Please recheck.
  • 3A,B,C (page 5) depicting inclusion structures are too small and should be magnified for better visibility, particularly host-guest interactions. Moreover, the intermolecular H-bond parameters of the three complexes should be tabulated and explicitly stated in text (page 4).
  • 3B shows the structure of G-Re-b-CD inclusion complex without H-atoms. Please recheck and correct.
  • Bands/peaks in Figs. 4-6 should be highlighted to clearly indicate the inclusion complex formation.
  • Characterizations by FTIR, DSC, XRD and SEM do not provide a direct evidence of the G-Re-g-CD inclusion complex formation. On the contrary, the chemical shift changes of CD internal protons H3 and H5 upon inclusion of a guest molecule in the CD cavity and the resulted host-guest interactions, indicating the inclusion complexation can be monitored and quantified (as binding constant) by 1H NMR spectroscopy. Hence, it is suggested to include the NMR results in the MS, otherwise further explanation and/or a comparison to other relevant ginsenoside-g-CD complexes should be given.
  • Page 12: How are the H-atoms of a-, b-, and g-CDs obtained for molecular docking? Details should be provided.
  • Page 12: The estimated uncertainties of the binding energies are required to judge the significant differences in the binding energies of the three complexes.
  1. The MS requires significant re-editing to improve English. Given examples are:
  • Too many typos and misused words, e.g.
  • Abstract and other pages: "G-Re powder" not "G-Re power"
  • Page 2: Lee [18] prepared "solution" not "conclusion"
  • Page 3: "Linearly was the AL-type's characteristic system" ... should be "Linearity in phase solubility is the characteristics of an AL-type system" ...
  • Page 5: "FT-IR spectroscopy is a convenient tool to verify …" appeared in two consecutive sentences
  • Page 7: "However" began in two consecutive sentences
  • Page 12, line -2: … a-CD, b-CD, and g-CD "molecules" not "modules"
  • Page 13: FTIR stands for "Fourier-transform infrared" not "Fourier-transformation infrared"

… and many more.

  1. The MS organization and preparation need serious improvement.
  • Abstract does not have a logical flow of key statements from all parts of the MS. It should be rewritten.
  • The six-line Conclusion is too short and does not cover the whole study.
  • Section 2. Results (pages 3-10) and Section 3. Discussion (mostly on page 11) should be merged for better understanding of the work. The one-page discussion does not refer to Figs. 1-10 and Table 1 at all.
  • Page 11: The paragraph discussing the molecular docking results is redundant and should be rewritten concisely.
  • 2-6 and 9 are too small to be seen clearly.
  • The values and unit of Ks are wrongly stated in the abstract and throughout the MS.
  1. References
  • The references partly conform to the journal format. Please check and correct.
  • Some relevant reviews on ginsenosides and articles on ginsenoside-CD complexes should be included in Introduction and/or Results and discussion.

Reviews

Shi, Z. Y., Zeng, J. Z., & Wong, A. S. T. (2019). Chemical structures and pharmacological profiles of ginseng saponins. Molecules, 24(13), 2443. 10.3390/molecules24132443

Peng, D., Wang, H., Qu, C., Xie, L., Wicks, S. M., & Xie, J. (2012). Ginsenoside Re: its chemistry, metabolism and pharmacokinetics. Chinese Medicine, 7(1), 1-6. 10.1186/1749-8546-7-2

Articles

Floresta, G., Punzo, F., & Rescifina, A. (2019). Supramolecular host-guest interactions of pseudoginsenoside F11 with β-and γ-cyclodextrin: Spectroscopic/spectrometric and computational studies. Journal of Molecular Structure, 1195, 387-394. 10.1016/j.molstruc.2019.05.134

Nam, Y. H., Le, H. T., Rodriguez, I., Kim, E. Y., Kim, K., Jeong, S. Y., ... & Kang, T. H. (2017). Enhanced antidiabetic efficacy and safety of compound K⁄ β-cyclodextrin inclusion complex in zebrafish. Journal of ginseng research, 41(1), 103-112. 10.1016/j.jgr.2016.08.007

Author Response

Response to Reviewer 1 Comments

Point 1: The complex binding energies from molecular docking calculation need to be rechecked and their derivation should be given; the value of G-Re-g-CD complex (6.70 kcal/mol) is highest compared to other two complexes (-4.7 and -5.1 kcal/mol), which is in contrast to the order of Ks of the three complexes. However, the authors conclude that both binding energies and stability constants have the same tendency. Please recheck.

Response 1: Thank you for your comment. Value of binding energies for G-Re and γ-CD complex was -6.7 kcal/mol. In the original version, we mistakenly wrote "- 6.7 kcal/mol " as "6.7 kcal/mol ", which has been modified in the revised version.

According to your suggestion, we have further investigated the principle of autodock software about binding energy calculation, and made a brief explanation, which was added to the revised manuscript, the content was as follows:

“In order to rationalize the mechanism of the inclusion process, molecular dynamics studies was performed. The Mark's genetic method was employed for conformation searches for 50 times. In auto dock, the binding energy G was calculated according to the formula as follows[22]: G=(Binding Energy)= G(Intermolecular Energy)+G(Internal Energy)+G(Torsional Energy) G(Torsional Energy)G(Unbound Extended Energy)”

Point 2: 3A, B, C (page 5) depicting inclusion structures are too small and should be magnified for better visibility, particularly host-guest interactions. Moreover, the intermolecular H-bond parameters of the three complexes should be tabulated and explicitly stated in text (page 4).

Response 2: The ABC in Figure 3 has been appropriately enlarged so that the hydrogen bond can be clearly displayed. In addition, we have added Table 2 to show the length of the hydrogen bond.

Table 2 Phase solubility parameters of G-Re in different CDs in water

CD

S0(×10-3M)

Slope

Ks(×103M-1)

1/slope

R2

△    G

(kJ/mol)

α-CD

0.392

0.0086

0.022

116.28

0.9938

-7.659

β-CD

0.391

0.1935

0.612

5.16

0.9933

-15.889

γ-CD

0.391

0.8494

14.41

1.18

0.9974

-23.736

*:“side”: the wide or the narrow side; “Insertion position”: the part include in the host molecule, Glc-C20 means glucose residue at C20

Point 3: 3B shows the structure of G-Re-β-CD inclusion complex without H-atoms. Please recheck and correct.

Response 3: Thank you for your suggestion. We modified Fig3 distinguished the original image by color, using red to represent oxygen atoms, and white to represent hydrogen atoms. Please refer to the modified manuscript.

Point 4: Bands/peaks in Figs. 4-6 should be highlighted to clearly indicate the inclusion complex formation

Response 4: We redrawn figure 4-6 and marked bands / peaks with color line segments or arrows for highlighting. Please refer to the modified manuscript.

Point 5: Characterizations by FTIR, DSC, XRD and SEM do not provide a direct evidence of the G-Re-g-CD inclusion complex formation. On the contrary, the chemical shift changes of CD internal protons H3 and H5 upon inclusion of a guest molecule in the CD cavity and the resulted host-guest interactions, indicating the inclusion complexation can be monitored and quantified (as binding constant) by 1H NMR spectroscopy. Hence, it is suggested to include the NMR results in the MS, otherwise further explanation and/or a comparison to other relevant ginsenoside-g-CD complexes should be given.

Response 5: Thank you for your comment. Nuclear magnetic technology is indeed an advanced and effective means to identify inclusion formation, and has been increasingly used in the preparation of inclusions in recent years. The main purpose of this research is to use r-cyclodextrin inclusion compound technology to improve the solubility and bioavailability of the poor-soluble compound G-Re. In order to achieve this goal, we first obtained the result of better compatibility between r-cyclodextrin and G-Re through phase solubility experiments, and further obtained the binding constants of the two through theoretical calculations, which proved that the inclusion process can be spontaneously. Next, the structure of the inclusion compound of G-Re and r cyclodextrin was verified by means of FTIR, DSC, XRD and SEM. Although the exact positions of the host molecule and the guest molecule cannot be obtained, the difference between the inclusion compound, the physical mixture and the monomer shows that the inclusion compound is different from the physical mixture, suggesting that the inclusion effect exists. In addition, there is a big difference in the dissolution between the inclusion compound and the physical mixture. The effective increase of the bioavailability of the inclusion compound is also a good proof of the formation of the inclusion compound. Many documents have verified the existence of inclusion compounds by the above four methods (Audino, PG Molecules 2005Ferrero, RMolecules 2021 et al). Above, we believe that the methods used in our experiments can support the experimental conclusions. Indeed, the nuclear magnetic identification experiment is indeed very helpful for enriching the content of our paper. Thank you again for your suggestions. We will carry out related experiments of nuclear magnetic identification in the follow-up work.

Point 6: Page 12: How are the H-atoms of a-, b-, and g-CDs obtained for molecular docking? Details should be provided.

Response 6:  Thank you for your comments. The details were added as follows:

If the distance and angle of two heteroatoms are within a certain range (generally considered to be <3.5 Å), it is considered that they can form a hydrogen bond interaction mediated by H atoms, that is, share a hydrogen atom to form a hydrogen bond

Point 7: Page 12: The estimated uncertainties of the binding energies are required to judge the significant differences in the binding energies of the three complexes.

Response 7: Thank you for your comments. The estimated uncertainties of the binding energies was Described in the revised manuscript:

Binding energies were calculated via the Lamarckian genetic algorithm and conformations that constitute each cluster were defined by a root mean square deviation (RMSD) tolerance of 2.0”.

Please refer to the revised manuscript.

Point 8: The MS requires significant re-editing to improve English. Given examples are:

  • Too many typos and misused words, e.g.
  • Abstract and other pages: "G-Re powder" not "G-Re power"

Response"G-Re power" was modified to "G-Re powder" in abstract and other pages.

  • Page 2: Lee [18] prepared "solution" not "conclusion"

Response"conclusion" was modified to "inclusion"

  • Page 3: "Linearly was the AL-type's characteristic system" ... should be "Linearity in phase solubility is the characteristics of an AL-type system".

ResponseThe sentence "Linearly was the AL-type's characteristic system" was modified to "Linearity in phase solubility is the characteristics of an AL-type system" .

  • Page 5: "FT-IR spectroscopy is a convenient tool to verify …" appeared in two consecutive sentences:

Response "FT-IR spectroscopy is a convenient tool to verify …" in second sentence was deleted.

  • Page 7: "However" began in two consecutive sentences

ResponsePage 7 the second "However" was deleted.

  • Page 12, line -2: … a-CD, b-CD, and g-CD "molecules" not "modules"

ResponseIn Page 12, line -2, "modules" were modified to "molecules ".

  • Page 13: FTIR stands for "Fourier-transform infrared" not "Fourier-transformation infrared"

Response"Fourier-transformation infrared “was modified to "Fourier-transform infrared"

Point 9:.The MS organization and preparation need serious improvement.

  • Abstract does not have a logical flow of key statements from all parts of the MS. It should be rewritten.

Response 9 Thank you for your comments. The abstract has been rewritten. The modified abstract was below:

"This study's purpose is to improve solubility Ginsenoside (G)-Re by forming inclusion complex. The solubility parameters of G-Re in alpha (α), beta (β), and gamma (γ) cyclodextrin (CD) were investigated. The phase solubility profiles were all classified as AL-type that indicated the 1:1 stoichiometric relationship with the stability constants Ks 22M-1 (α-CD), 612 M-1 (β-CD) and 14410 M-1 (γ-CD) respectively. Molecular docking studies confirmed the results of phase solubility with the binding energy were −4.7 (α-CD) −5.10 (β-CD) and -6.70 (γ-CD) kcal/mol respectively. The inclusion complex (IC) of G-Re was prepared with γ-CD via the water-stirring method followed by freeze-drying. The successful preparation of IC was confirmed by power X-ray diffraction (XRD), Fourier transform-infrared spectroscopy (FT-IR), differential scanning calorimetry (DSC), and scanning electron microscopy (SEM). In-vivo absorption studies were carried out by LC-MS/MS. Dissolution rate of G-Re was increased 9.27 times after inclusion, and the peak blood concentration was 2.7-fold higher than that for pure G-Re powder. The relative bioavailability, calculated from the ratio of AUC0-∞ of the inclusion to pure G-Re power was 171%. Current study offers the first report that describes G-Re's inclusion into γ-CD, and explored the inclusion complex's mechanism from the molecular level. The result indicated that the solubility could be significantly improved as well as the bioavailability. γ-CD was a very suitable inclusion host for complex preparation of G-Re".

Point 10: The six-line Conclusion is too short and does not cover the whole study.

Response 10Thank you for your comments. The Conclusion has been rewritten. The written summary was below. Please refer to the revised manuscript.

"The host-guest supramolecular interactions between α-CD, β-CD, γ-CD and G-Re have been investigated in this manuscript. All experiments have confirmed that G-Re and γ-CD form a 1:1 inclusion compound. Affected by the size of the guest molecule of G-Re, the host molecule of γ-CD can only form a partial IC with G-Re at a molar ratio of 1:1. Molecular docking studies elaborated the complex mechanism that as a protopanaxatriol G-Re can only be partially embedded in the cavity of γ-CD due to the restriction of sugar chains. However, even if a partial IC was formed, the IC of G-Re-γ-CD would also increase the dissolution rate and bioavailability. The effect of γ-CD on the enhancement of G-Re’s dissolution and bioavailability was significant.

An inclusion complex with G-Re and γ-CD with a molar ratio of 1:1 was further prepared. The characteristics of G-Re were different from those in the physical mix-ture and inclusion compound. The successful preparation of the G-Re-γ-CD IC was confirmed by FT-IR, DSC, XRD, and SEM. The dissolution rate of G-Re from the inclusion was greatly improved. The G-Re amount released from IC was 10.27-times greater than that for G-Re power. Therefore, the dissolution rate could be greatly improved with γ-CD inclusion. In vivo studies after oral administration showed that the AUC value of γ-CD IC was higher than that of pure G-Re. After oral ad-ministration of G-Re-γ-CD IC at a dose of 100 mg/kg the peak concentrations (Cmax) and the bioavailability of G-Re was 1.71-times higher than the pure G-Re powder power. Therefore γ-CD is a very suitable inclusion host for complex preparation of G-Re".

Point 11: Section 2. Results (pages 3-10) and Section 3. Discussion (mostly on page 11) should be merged for better understanding of the work. The one-page discussion does not refer to Figs. 1-10 and Table 1 at all.

Response 11Thank you for your comments. The discussion part was merged into Results,and the title of part section 2 " Results " was modified to " Results  and discussion". And the content was appropriately supplemented (marked red) to refer to Figs. 1-10 and Table 1. Please referred to the revised manuscript.

Point 12: Page 11: The paragraph discussing the molecular docking results is redundant and should be rewritten concisely.

Response 12The paragraph discussing the molecular docking results was simplified and combined with the result part, the modified part was as follows:

"In order to rationalize the mechanism of the inclusion process, molecular mod-eling studies was performed. The Mark's genetic method was employed for conformation searches for 50 times. In auto dock, the binding energy G was calculated according to the formula as follows[22]: G=(Binding Energy)= G(Intermolecular Energy)+G(Internal Energy)+G(Torsional Energy) G(Torsional Energy)G(Unbound Extended Energy). Mark's genetic method was used to perform conformation searches for 50 times, The calculated binding energies given after calculation were −4.7 kcal /mol (α-CD), −5.10 kcal /mol (β-CD), and −6.70 kcal /mol (γ-CD) respectively. For the lower binding energy, the more stable the conformation were formed [23], the structure of G-Re-γ-CD inclusion complex was more stable than the other two, which was consisted with the result of phase solubility study.

The most stable conformations of G-Re-CDs inclusion complexes were shown in Figure 3. In the conformations of the three inclusion compounds, G-Re all extended from the wide side to the narrow side of the cyclodextrin. However, affected by the cavity size, the binding ways of G-Re to CDs were different: When G-Re co-existed with α-CD, the glucose residue at position C20 inserted into the cavity from wide side with six hydrogen bonds formatted with the oxygen and hydrogen atoms with the hydrogen bond lengths were (2.5, 2.8, 2.4, 2.5, 2.5, 2.3 Å)(Figure 3.A). Therefore, for the complex of G-Re-α-CD, hydrogen bonding was the main forces to maintain stability. When G-Re encountered with β-CD, G-Re inserted its glucose residue at C6 into the cavity also from the wide side with seven hydrogen bonds formatted with the oxygen on glucose residue and hydrogen on β-CD (3.3, 3.1, 3.1, 2.9, 3.2, 2.8, 3.2 Å)(Figure 3.B). The cavity size of β-CD is larger than that of α-CD, so there is more space that can accommodate the guest. The β-CD provided a more stable conformation for the guest, and on the other hand, it can also provide more opportunities for the formation of hydrogen bonds. Therefore, the theoretical calculation results showed that the binding energy of G-Re-β-CD was lower than that of α-CD, which was also consistent with the calculation results of the Ks. γ-CD is a cavity structure composed of eight D-(+)-glucopyranose monomers with the cavity size of 7.5-8.3 Å. As seen from Figure 3 C, γ-CD could provide enough space to chain of the glucose residue at C20 passes through its cavity. Four hydrogen atoms (1.9, 2.1, 2.0, 2.4 Å) were formed of the hydroxyl group on the glucose residue at C20 and C6 participated in the formation of hydrogen bonds from the wide side and the narrow side respectively, which provided a force for the stability of the inclusion.

Molecular docking studies were adopted to elaborate the complex mechanism of host and guest molecule [24]. The structures of host molecule-CDs were fairly rigid due to intra molecular hydrogen bonds between the O-2H and the O-3H of adjacent glucose units [25]. Therefore, a semi-flexible docking form was adopted for the simulation of G-Re and CDs. The docking calculation results provided an interpretation of the solubility test in the formation of inclusion compounds between G-Re and CDs at the molecular level. From a structural point of view, this stable combination was attributed to the larger cavity volume of γ-CD and the more stable spatial conformation formed by the guest and the host. Docking calculation results also provided theoretical support for the 1:1 molar ratio inclusion complex inferred by the phase solubility. For the molecule size, only the glycoside residue of G-Re can extend or partially extend into the cavity. The most stable conformation of the non-inclusion structure was parallel to the wide side of the CD. In this way, G-Re will not have enough space to combine with other cyclodextrin molecules. Therefore, only a 1:1 molar ratio inclusion complex could be conformed. The calculated results of Docking further confirm the calculated results of the solubility constant from a microscopic point of view. According to the calculation of docking calculation, the difference of the three inclusion constants may be due to the differences of cavity sizes. γ-CD could include the whole sugar residue at C20 through the cavity, and this conformation was conducive to the formation of hydrogen bonds between the hydroxyl groups on both sides of the wide and small segments with CDs. "

Point 13: 2-6 and 9 are too small to be seen clearly.

Response 13:Figure 2-6 and 9 were properly modified to make the picture seen clearly.

Point 14: The values and unit of Ks are wrongly stated in the abstract and throughout the MS.

Response 14Thank you for your comments. The unit of Ks should be “M-1”, Three places were involved in the manuscript, and they have been modified separately, please referred to the revised manuscript.

Point 15: The references partly conform to the journal format. Please check and correct.

Response 15All reference formats have been revised to meet the requirements of this journal.

Point 16: Some relevant reviews on ginsenosides and articles on ginsenoside-CD complexes should be included in Introduction and/or Results and discussion.

Response16Related references have been added to the revised manuscript. The numbers of the supplementary references are 1, 8, 18 and 20, which are marked red. Please refer to the modified manuscript. Thank you for your comments again.

Reviewer 2 Report

The article ”Preparation, characterization, and bioavailability of host‐guest inclusion complex of ginsenoside Re with gamma‐cyclodextrin” has a well-organized structure, the research seems to be conducted well. As the authors mentioned in the introduction section, there are many studies on native cyclodextrin with ginsenoside component, so the idea’s originality stays in ginsenoside Re.

I suggest that some corrections should be done:

Minor:

1, Figure 2. has poor quality and the standard deviation is missing from the curves. Please edit the attached table based on the molecules’ template.

2, In section 2.4 Preparation and optimization of IC, “Equal amounts of G‐Re and γ‐CD were shaken at 30°C for 30 minutes to prepare a physical mixture (PM) for comparison”. Please give more details on how to prepare the physical mixture (solvent, freeze dryer).

3, Some figure labels contain mistakes in the notation of the compositions: G‐Re (A), γ‐CD (B), G‐Re‐γ‐CD PM (B), and G‐Re‐γ‐CD IC (D)- please correct to G‐Re (A), γ‐CD (B), G‐Re‐γ‐CD PM (C), and G‐Re‐γ‐CD IC (D).

4, Figure 7. has poor quality and the scale bar is invisible.

Major:

1, To increase the quality of the manuscript, it would be necessary to perform an in vitro permeability study on a Caco-2 cell line.

2, Authors need to pay more attention to improving the discussion, especially the in vivo results.

Author Response

Response to Reviewer 2 Comments

Point 1Figure 2. has poor quality and the standard deviation is missing from the curves. Please edit the attached table based on the molecules’ template.

Response 1: Thank you for your comment. Figure 2 has been modified, the standard deviation was highlighted, and the calculated parameters were reorganized into Table 1. Please refer to the revised manuscript.

Point 2 In section 2.4 Preparation and optimization of IC, “Equal amounts of G‐Re and γ‐CD were shaken at 30°C for 30 minutes to prepare a physical mixture (PM) for comparison”. Please give more details on how to prepare the physical mixture (solvent, freeze dryer).

Response 2: Thank you for your comment. A physical mixture (PM) refers to a mixture prepared by physically mixing host molecules and guest molecules. This process does not require a dissolution process. Therefore it does not involve dissolution and freeze-drying processes. The detailed description of the preparation process of the physical mixture is added as

"A physical mixture (PM) refers to a mixture prepared by physically mixing host molecules and guest molecules. Equal amounts of G‐Re and γ‐CD were shaken at 30°C for 30 minutes to prepare a physical mixture for comparison."

Point 3Some figure labels contain mistakes in the notation of the compositions: G‐Re (A), γ‐CD (B), G‐Re‐γ‐CD PM (B), and G‐Re‐γ‐CD IC (D)- please correct to G‐Re (A), γ‐CD (B), G‐Re‐γ‐CD PM (C), and G‐Re‐γ‐CD IC (D).

Response 3: Figure labels of "G‐Re (A), γ‐CD (B), G‐Re‐γ‐CD PM (B), and G‐Re‐γ‐CD IC (D)" were modified to "G‐Re (A), γ‐CD (B), G‐Re‐γ‐CD PM (C), and G‐Re‐γ‐CD IC (D)", please refer to the modified manuscript.

Point 4Figure 7. has poor quality and the scale bar is invisible.

Response 4: We have modified Figure 7 properly so that the scale bar can display normally,please refer to the modified manuscript.

Major:

Point 5To increase the quality of the manuscript, it would be necessary to perform an in vitro permeability study on a Caco-2 cell line.

Response 5: Thank you for your comment. The main purpose of our experiments is to confirm the inclusion complex can increase the absorption of guest molecule G-Re. This was well demonstrated by the increased bioavailability in rats. Caco-2 cell model is a human clonal colon adenocarcinoma cell which can be used to simulate intestinal transport mechanism in vivo. Adding the Caco-2 cell experimental model can indeed further improve the quality of the manuscript, however the time for revising the draft is limited to complete the exploration of the cell experimental mechanism. The results of in vivo test can prove that the bioavailability has been improved. Further, we planed to explore the mechanism of increased bioavailability through in vitro permeability study on a Caco-2 cell line in next experiment. Thank you again for your comment.

Point 6Authors need to pay more attention to improving the discussion, especially the in vivo results.

Response 6: The discussion in each part has been implemented, and the discussion part is written together with the results part. Particularly, the supplementary part for the in vivo test results is as follows:

"Compared with the group administered G-Re powder, the Tmax of G-Re-γ-CD IC was shorter, the AUC0-24h increased and the bioavailability was improved. On the one hand, the increased in the G-Re content in the body in a molecular state caused more G-Re molecules to penetrate the mucus layer and the immobile water layer. And the contact surface of the G-Re cell membrane increases. This result suggested that increased the solubility of G-Re can significantly increase its blood access. On the other hand, the oil-water partition coefficient of G-Re-γ-CD IC was larger than that of G-Re powder [36, 37]. It is speculated that G-Re-γ-CD maintains the coexistence balance of inclusion compound and non-inclusion complex, and plays a transfer role at the water-biological phase interface, which increases the amount of G-Re permeating through the biofilm, thereby making improved bioavailability. The increased G-Re content in the blood takes longer to clear, so T1/2 of G-Re-γ-CD IC was prolonged. Additionally, related studies have shown that cyclodextrin and its derivatives can deplete cholesterol on cell membranes, change their structure and permeability, thereby inhibiting the activity of P-glycoprotein, which is closely related to the bioavailability of oral drugs [38]. These properties of dextrin can increase the probability of the drug entering the intestinal epithelial cells, thereby prolonging the drug MRT(h) and T1/2(h). And the inclusion compound can also improve the stability of the drug in body fluids and increase its absorption [39]. This also suggests that the preparation of G-Re into G-Re-γ-CD inclusion compound can improve its oral bioavailability. And above research content has important reference significance for the development of G-Re and related pharmaceutical dosage forms. "

Round 2

Reviewer 1 Report

Authors have addressed most of the questions/comments and the revised manuscript has been significantly improved both presentation and organization. More importantly, without a direct evidence of inclusion formation from 1H NMR data, it is unconvincing that the inclusion modes and host-guest interactions proposed by authors using molecular docking are plausible (Fig. 3A,B,C and Table 2), although the results from FT-IR, XRD and DSC indicate the G-Re-g-CD complex formation. Authors have to state this limitation in the revised manuscript.

Moreover, a number of points about English language are still overlooked and writing style should be improved. Hence, English editing is required before proceeding with publication in Molecules. Listed below are just some obvious examples.

  • P1, L10, Abstract: "This study's purpose is to improve solubility Ginsenoside ..." should be "This work aims at improving the water solubility of Ginsenoside ..."
  • P2, L59-60: "Significantly improve ..." This is not a sentence. Please correct.
  • P2, L64-66: "So can compounds with a relatively large ...?" Two question marks are missing from the first two questions.
  • P2, L71: "Inclusion complex formation" An apostrophe +"s" ('s) is not needed.
  • P3, Table 1: The stability constants (Ks) of a-, b- and g-CDs complex with G-Re are 22, 612 and 14410 M-1, respectively. Because the values of Ks are rather varied (22-14410 M-1), they should be given with their estimated standard errors to indicate the significances of both Ks values and their differences.
  • P3, L97-99: "a 1:1 molar ratio" appeared twice in one sentence. This should be rewritten.
  • P4, L126: "The calculated binding energies given after calculation ..." The redundant words "calculated" and "calculation" should be avoided in a sentence.
  • P5, Table 2: "Bond lengths" should be more precisely written as "O...O intermolecular distance"
  • P6, Fig. 3: The intermolecular O-H…O H-bonds should be shown clearly.
  • P9, L258-260: "However" began in two consecutive sentences. This point has not been corrected by authors in the revision.
  • P18, L563-564: The word "powder" was repeated twice.

Author Response

Response to Reviewer 1 Comments 2

Point 1Authors have addressed most of the questions/comments and the revised manuscript has been significantly improved both presentation and organization. More importantly, without a direct evidence of inclusion formation from 1H NMR data, it is nconvincing that the inclusion modes and host-guest interactions proposed by authors using molecular docking are plausible (Fig. 3A,B,C and Table 2), although the results from FT-IR, XRD and DSC indicate the G-Re-g-CD complex formation. Authors have to state this limitation in the revised manuscript.

Response 1: Thank you for your comment. The limitation has been stated in the revised manuscript as that “However, the hydrogen bond generated by the combination of host molecule (cyclodextrin) and guest molecule (G-Re) is the result of molecular dynamics simulation. A more accurate combination mode needs to be further verified by NMR experiments.” Please refer to the revised manuscript in section 2.2.

Point 2Moreover, a number of points about English language are still overlooked and writing style should be improved. Hence, English editing is required before proceeding with publication in Molecules. Listed below are just some obvious examples.

Response 2: Thank you for your comment. The English writing of the full text has been polished and revised. The changes were marked in red. Please refer to the revised manuscript.

  • P1, L10, Abstract: "This study's purpose is to improve solubility Ginsenoside ..." should be "This work aims at improving the water solubility of Ginsenoside ..."

ResponseThe sentence was modified to “This work aims at improving the water solubility of Ginsenoside ...”

  • P2, L59-60: "Significantly improve ..." This is not a sentence. Please correct.

ResponseThe sentence " Lee [21] prepared inclusion with intestinal metabolite of ginseng saponin (IH901) and β-CD. Significantly improve the solubility and bioavailability of intestinal metabolites of ginsenosides (IH901) with β-CD. ..." in P2, L59-60 was modified to “And when combined with β-CD, a remarkable improvement of the solubility and bioavailability of intestinal metabolite of ginseng saponin (IH901) was shown

  • P2, L64-66: "So can compounds with a relatively large ...?" Two question marks are missing from the first two questions.

ResponseThe sentence " So can compounds with a relatively large ...? " in P2, L64-66 was modified to " So, can compounds with a relatively large molecular diameter like G-Re form IC with CD? What kind of conformation will be formed? And how much will the dissolution and bioavailability of the IC be improved?"

  • P2, L71: "Inclusion complex formation" An apostrophe +"s" ('s) is not needed.

ResponseThe sentence “Inclusion complex formation” in P2, L71 was modified to “Moreover, inclusion complex formation was identified by chemical and physical means

  • P3, Table 1: The stability constants (Ks) of a-, b- and g-CDs complex with G-Re are 22, 612 and 14410 M-1, respectively. Because the values of Ks are rather varied (22-14410 M-1), they should be given with their estimated standard errors to indicate the significances of both Ks values and their differences.

ResponseThank you for your comment. We measured the solubility values of G-Re in  cyclodextrin solutions with different concentrations for three times in parallel, and calculated the slope after averaging them, later obtained the Ks value based on the slope and S0. Therefore, Ks is not obtained by the average of three calculations, but calculated by the slope.

  • P3, L97-99: "a 1:1 molar ratio" appeared twice in one sentence. This should be rewritten.

ResponseIn P3, L97 “a 1:1 molar ratio” was deleted.

  • P4, L126: "The calculated binding energies given after calculation ..." The redundant words "calculated" and "calculation" should be avoided in a sentence.

ResponseThe sentence “The calculated binding energies given after calculation ...” in P4, L126 was modified to “the calculated binding energies were −4.70 kcal /mol (α-CD), −5.10 kcal /mol (β-CD), and −6.70 kcal /mol (γ-CD) respectively.

  • P5, Table 2: "Bond lengths" should be more precisely written as "O...O intermolecular distance"

Response "Bond lengths" in P5, Table 2 were modified to “O-O intermolecular distance

  • P6, Fig. 3: The intermolecular O-H…O H-bonds should be shown clearly.

ResponseThank you for your comment. In the last revision of the document, we modified the representation in Figure 3, represent the chimeric mode between the guest molecule and the host molecule, and the hydrogen bonds are presented in the form of Table 3.

  • P9, L258-260: "However" began in two consecutive sentences. This point has not been corrected by authors in the revision.

ResponseThe sentence “Crystals usually have a distinct melting point (Tm). However the amorphous form often takes the glass transition temperature (Tg) as the characteristic thermodynamic parameter” in P9, L258-260 was modified to “Crystals usually have a distinct melting point (Tm), yet the amorphous form often takes the glass transition temperature (Tg) as the characteristic thermodynamic parameter”.

  • P18, L563-564: The word "powder" was repeated twice.

ResponseIn P18, L563-564, the second word “powder” was deleted.

Reviewer 2 Report

The authors have satisfactorily responded to all my questions.

Author Response

Response to Reviewer 2 Comments 2

no Comments